# An in-silico approach to predict and exploit synthetic lethality in cancer metabolism

Iñigo Apaolaza [1], Edurne San José-Eneriz[2], Luis Tobalina [1,3], Estíbaliz Miranda[2], Leire Garate[2], Xabier Agirre[2], Felipe Prósper[2] & Francisco J. Planes[1]

Synthetic lethality is a promising concept in cancer research, potentially opening new possibilities for the development of more effective and selective treatments. Here, we present a computational method to predict and exploit synthetic lethality in cancer metabolism. Our approach relies on the concept of genetic minimal cut sets and gene expression data, demonstrating a superior performance to previous approaches predicting metabolic vulnerabilities in cancer. Our genetic minimal cut set computational framework is applied to evaluate the lethality of ribonucleotide reductase catalytic subunit M1 (RRM1) inhibition in multiple myeloma. We present a computational and experimental study of the effect of RRM1 inhibition in four multiple myeloma cell lines. In addition, using publicly available genome-scale loss-of-function screens, a possible mechanism by which the inhibition of RRM1 is effective in cancer is established. Overall, our approach shows promising results and lays the foundation to build a novel family of algorithms to target metabolism in cancer.

[1] CEIT and Tecnun, University of Navarra, Manuel de Lardizábal 15, 20018 San Sebastián, Spain. [2] Area de Hemato-Oncología, IDISNA, Ciberonc, Centro de Investigación Médica Aplicada (CIMA), University of Navarra, Pío XII 55, 31008 Pamplona, Spain. [3] Faculty of Medicine, Joint Research Centre for Computational Biomedicine, RWTH Aachen University, MTI2 Wendlingweg 2, D-52074 Aachen, Germany. Iñigo Apaolaza and Edurne San José-Eneriz contributed equally to this work. Correspondence and requests for materials should be addressed to F.P. (email: fprosper@unav.es) or to F.J.P. (email: fplanes@tecnun.es)

Synthetic lethality is defined as a type of genetic interaction where the co-occurrence of two (or more) genetic events results in cellular death, while the occurrence of either event on its own is compatible with cell viability[1]. Given the underlying genetic variations in tumor cells, synthetic lethality is a promising approach in cancer research as it largely expands the number of possible drug targets and creates an opportunity for selectivity[2]. The increasing evidence of metabolic reprogramming of cancer cells makes it ideal to exploit the concept of synthetic lethality, as illustrated in different previous works[3, 4].

In order to accelerate the pace of discovery, a number of in silico tools have been developed to predict metabolic targets in cancer. In particular, constraint-based modeling (CBM) for genome-scale metabolic networks, which takes into account mass balance and thermodynamic constraints as well as available –omics data, has received much attention[5–8]. CBM methods first contextualize a reference metabolic network for the type of cancer under study using –omics data and, subsequently, computationally predict growth rate under gene knockout perturbations (gene essentiality analysis[9]). Potentially effective therapeutic strategies are those gene knockouts sufficiently restricting the growth rate, a key phenotype to be disrupted in cancer.

The step of building cancer-specific metabolic networks is crucial for these approaches and strongly biases the results of gene essentiality analysis. Previous studies have found conflicting results between existing algorithms[7, 10], as they depend on different heuristic choices to integrate–omics data. In addition, this step masks the concept of synthetic lethality, because it suppresses (at least part of) repressed alternative pathways for biomass production that may explain the essentiality of a particular gene in cancer.

To avoid these issues, we propose here to directly calculate knockout strategies on the reference (uncontextualized) human metabolic network and select the most lethal ones in cancer using available–omics data, avoiding, in consequence, the step of cancer-specific metabolic reconstruction. To that end, we rely on the concept of minimal cut sets (MCSs), which defines minimal sets of reactions whose removal would render the functioning of a given metabolic task impossible, in our case the biomass reaction[11, 12]. Importantly, in order to predict synthetic lethality in cancer, we extend the concept of MCSs to the gene level (genetic minimal cut sets, gMCSs).

MCSs are suitable to carry out the analysis proposed above since they have nice mathematical properties that have been exploited for their more efficient computation even in large networks[13–15]. The advances achieved in MCS computation allow us to go beyond approaches that exhaustively analyze all possible combinations of gene/reaction knockouts, which are restricted to the identification of lower order synthetic lethals.

Here, we adapt a recent approach presented in Tobalina et al.[15], in order to (i) calculate gMCSs and (ii) use gene expression data as the main criterion to guide the search. Our approach allows us to exploit the concept of synthetic lethality and evaluate whether and why a particular gene knockout is lethal in cancer, showing a more accurate and informative performance than current methods in the literature.

Our gMCS computational framework is applied to evaluate the lethality of *ribonucleotide reductase catalytic subunit M1* (*RRM1*) in multiple myeloma (MM), a hematological cancer that remains for the most part an incurable disease. *RRM1* has been previously spotted as a promising metabolic target in different cancer studies[16–18]; however, a more systematic and unbiased analysis in MM was still lacking. Here, we present a computational and experimental analysis of the lethality of *RRM1* in four different MM cell lines. In addition, using publicly available genome-scale loss-of-function screens, we establish a possible mechanism by which the inhibition of *RRM1* is effective in cancer. The results obtained are promising, laying the foundation to build a novel family of algorithms to investigate and target cancer metabolism.

## Results

**Genetic minimal cut sets and cancer-specific essential genes.** Minimal cut sets at the gene level, referred here to as gMCSs, are minimal subsets of genes whose simultaneous removal directly blocks a particular metabolic task. In cancer studies, this target metabolic task is typically the biomass reaction, whose flux represents the proliferation rate, a key phenotype to disrupt in cancer.

Our approach avoids the step of building cancer-specific metabolic networks and determines gMCSs for biomass production on a reference human genome-scale metabolic network. In addition, using cancer gene expression data as source of evidence for reaction activity, we can exploit synthetic lethality and identify cancer-specific essential genes. In particular, we search for gMCSs that can be targeted using a single gene knockout because the rest of the genes comprising the gMCSs are lowly expressed.

Our approach is illustrated in Fig. 1. Assuming the example metabolic network shown in Fig. 1a and one-to-one gene-reaction

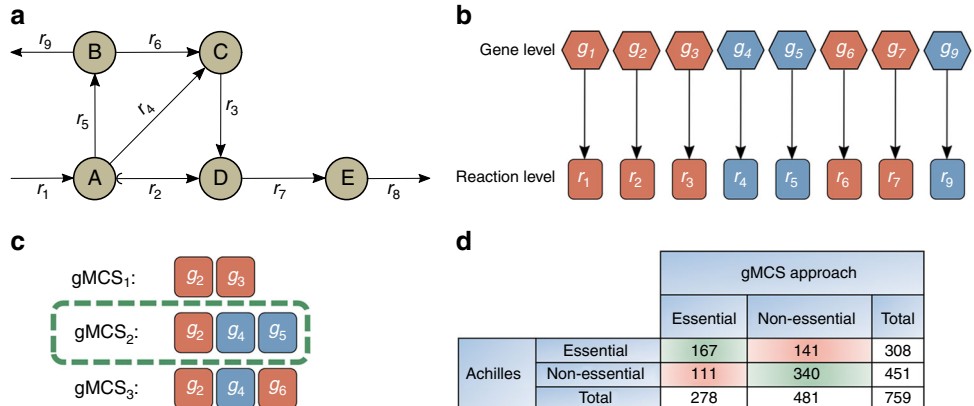

**Fig. 1** Identification of cancer-specific essential genes using our gMCS approach. **a** Example metabolic network. The target reaction is $r_8$. **b** Trivial gene–protein–reaction (GPR) rules scenario for **a**. *Blue coloring* is associated with lowly expressed genes, $L = \{4, 5, 9\}$, and *red coloring* is associated with moderately/highly expressed genes or reactions lacking gene annotation, $\bar{L} = \{1, 2, 3, 6, 7\}$. **c** Resulting gMCSs involving $g_2$ given the GPR rules in **b**. gMCS$_2$ explains the essentiality of $g_2$. **d** Contingency table with the essentiality predictions of the gMCS approach in the Project Achilles[19] data

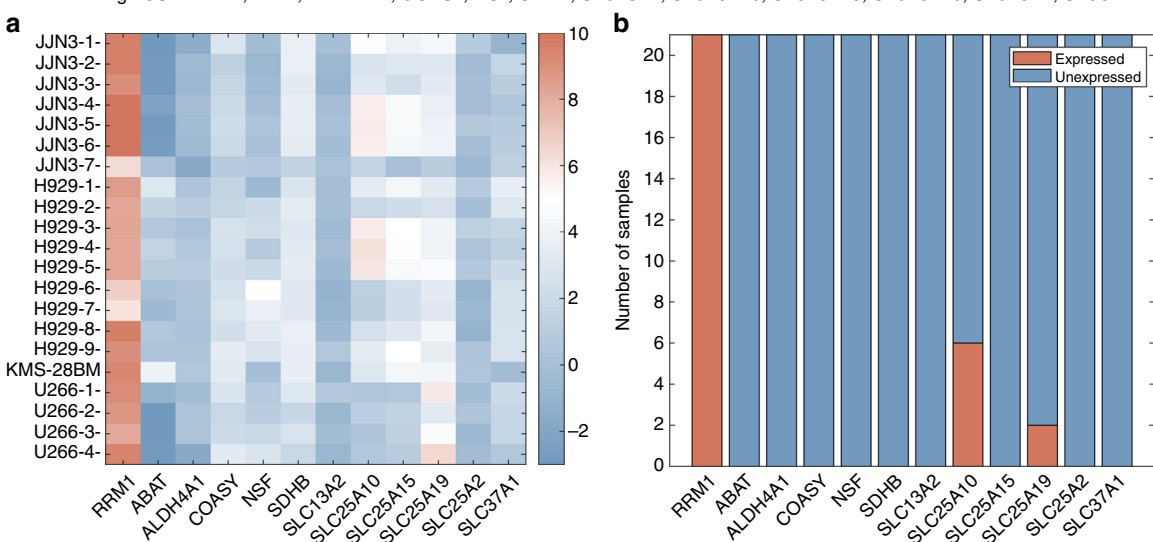

**Fig. 2** Expression levels of genes included in gMCS$_6$ for different samples of the MM cell lines considered. **a** Heatmap of Barcode $z$-scores[23] of *RRM1* and its partner genes (gMCS$_6$) in different MM samples analyzed. **b** Number of MM samples where *RRM1* and its partner genes (gMCS$_6$) are expressed/unexpressed, according to Barcode threshold of expression ($z \geq 5$)

relationships (GPR rules) (Fig. 1b), we find a gMCS (gMCS$_2$) involving genes 2, 4 and 5 ($g_2$, $g_4$ and $g_5$) (Fig. 1c). This gMCS explains here the essentiality of $g_2$ for producing metabolite $E$, given that $g_4$ and $g_5$ are not active in this particular context. Full details as to how to identify this type of gMCSs can be found in "Methods" section. A more detailed toy example illustrating the novelty brought by our method can be found in Supplementary Note 1. We describe the need to move from MCSs to gMCSs, showing that minimal subsets of reactions which block a given metabolic task are not necessarily minimal knockout strategies at the gene level as a consequence of non-trivial GPR rules.

Our main hypothesis is, thus, that the essentiality of a gene can be predicted by finding a gMCS where all its partner genes are lowly expressed, as in Fig. 1c. In order to provide a large-scale validation to our hypothesis, we used genome-scale loss-of-function screens provided by the Project Achilles v2.4.3[19]. Given the underlying noise in these experiments, we focused on the 30 cell lines with the highest quality score calculated as done in Hart et al.[20] and with expression data available in the Cancer Cell Line Encyclopedia[21] (Supplementary Table 1). For each cell line, we derived the top 20 most and least essential metabolic genes, according to the essentiality score calculated in Hart et al.[20], which permits to rank the genes according to their essentiality within a cell line.

A 5 min time limit was set to our algorithm for the calculation of gMCSs for each of the 600 Achilles-based essentiality and 600 Achilles-based non-essentiality cases (20 essential and 20 non-essential genes for each of the 30 cell lines considered). We used Recon2.v04 as the reference network[22] and RPMI1640 culture medium[6]. In addition, for each cell line, the sets of lowly expressed genes and reactions ($L$) were obtained following the Gene Expression Barcode 3.0 algorithm using standard parameters[23] and the Gene–protein-reaction rules provided by Recon2.v04 (see "Methods" section).

Under these conditions, our algorithm found a solution for 308 out of 600 Achilles-based essentiality cases within the time limit, predicting as essential 167 of them and incorrectly categorizing 141. Among the Achilles-based non-essentiality cases, on the other hand, out of 451 solutions found within the time limit, 340 were correctly identified as non-essential and 111

were incorrectly identified as essential. The results are summarized in Fig. 1d. A logistic regression was used to check whether more Achilles-based essentiality cases are predicted to be essential by our algorithm than Achilles-based non-essentiality ones, and the results are highly significant ($p$-value $= 3.78 \times 10^{-16}$, odds ratio (OR) $= 3.62$) (see "Methods" section).

These results were compared with standard gene essentiality analysis on metabolic reconstructions conducted with two well-known algorithms: GIMME[24] and iMAT[25]. The aim of reconstruction methods is to identify a subset of reactions from the reference metabolic network that best fits to available expression data and satisfies steady-state condition, thermodynamic constraints and biomass production. Once the reference network is contextualized using gene expression data, gene essentiality analysis is conducted, i.e., identification of single gene knockouts that disrupt biomass production. In this analysis we used the same input data as in our gMCS approach: Recon2.v04 and gene expression levels for cancer cell lines considered above. Full details about the implementation of GIMME and iMAT can be found in Supplementary Notes 1 and 2.

Out of the 600 Achilles-based essentiality cases, 49 were identified as essential by GIMME and 35 by iMAT. On the other hand, out of the 600 Achilles-based non-essentiality cases, 576 and 571 were identified as non-essential by GIMME and iMAT, respectively. As done with our gMCSs approach, we used a logistic regression to provide statistical significance to the aforementioned results: GIMME: $p$-value $= 0.003$, OR $= 2.13$; iMAT: $p$-value $= 0.44$, OR $= 1.22$ (Supplementary Tables 2 and 3).

It is remarkable that the gain in sensitivity brought by our algorithm while keeping a similar precision value (0.6–0.7). This is also observed with the statistical significance analysis of the logistic regression, which validates our hypothesis about essential genes and gMCSs discussed above. It is noteworthy to mention the significant number of potential cancer-specific essential genes obtained (in different cell lines) that are in agreement with Achilles data (Supplementary Data 1). They constitute an attractive subset of genes to explore in the future.

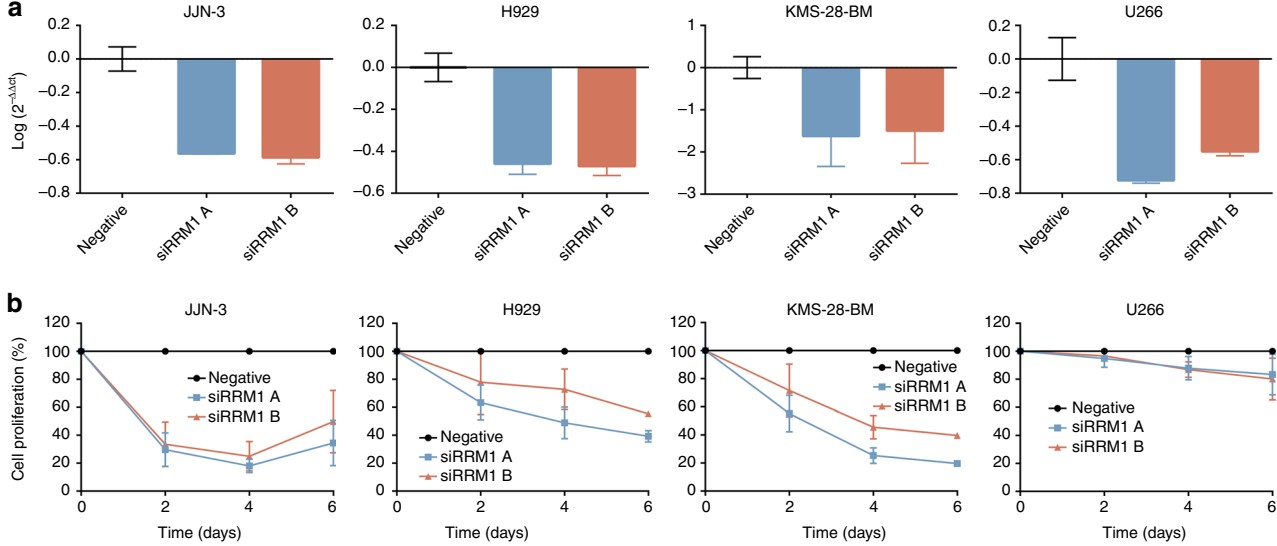

**Fig. 3** Gene silencing analysis of *RRM1* in JJN-3, H929, KMS-28-BM and U266 MM cell lines. **a** mRNA expression of *RRM1* gene 48 h after nucleofection with the specific siRNAs. Data are referred to *GUS* gene and an experimental group nucleofected with negative control siRNA. **b** Proliferation of JJN-3, H929, KMS-28-BM and U266 cell lines nucleofected with siRNAs targeted to *RRM1* gene was studied by MTS. The proliferation percentage refers to cells nucleofected with a negative control siRNA. Data represent mean ± standard deviation of at least three experiments

**RRM1 essentiality in MM**. Previous research has shown that *RRM1* plays an important role in cancer[16, 18]. *RRM1* constitutes the large regulatory subunit of the enzyme *ribonucleotide reductase*, which catalyzes the conversion of ribonucleoside diphosphates into deoxyribonucleoside diphosphates. *RRM1* binds to *RRM2* or *RRM2B* to conduct metabolic activity[17]. The expression of *RRM1* is increased in some tumor types and seems to correlate with cell proliferation[26]. Early promising results were found in different MM cell lines using an *RRM1* inhibitor didox (3,4-dihydroxybenzohydroxamic acid)[27]; however, several studies have shown the possible interaction of didox with additional targets aside from *RRM1*[28]. Here, using our predictive algorithm presented above, we present a systematic and unbiased analysis of the role of *RRM1* in four different MM cell lines: JJN-3, H929, KMS-28-BM and U266. We also provide in vitro siRNA silencing experiments in the same four MM cell lines.

gMCSs associated to *RRM1* which block proliferation (i.e., the biomass reaction) were computed using Recon2.v04 as the reference network[22] and RPMI1640 as culture medium[6]. Note that GPR rules for reactions involving *RRM1* were modified according to revised literature (Supplementary Note 3). We calculated up to three gMCS for seven samples of JJN-3, nine samples of H929, one sample of KMS-28-BM, and four samples of U266 (Supplementary Table 4). For each sample the sets of lowly expressed genes and reactions (*L*) were obtained following the Gene Expression Barcode 3.0 algorithm using standard parameters[23] and the Gene–protein-reaction rules provided by Recon2.v04. Overall, after removing repeats and cases not reaching feasible solution within the time limit (see "Methods" section), we obtained 20 gMCSs ($gMCS_1$–$gMCS_{20}$). It is important to emphasize that gMCSs are inherent to the target reaction and the reference network they have been calculated from (in our case Recon2.v04). For this reason, computed gMCSs can be mapped onto the gene expression data of all samples considered. Full details can be found in Supplementary Data 1.

Figure 2a shows a heatmap that represents the Barcode expression *z*-scores[23] of the genes involved in one of the 20 gMCSs calculated in the different samples analyzed, i.e., $gMCS_6$. All genes involved in this gMCS show a low expression in the majority of the samples (13 out of 21), except for *RRM1* which is highly expressed in all of them (Fig. 2b). In other words, this gMCS successfully explains the essentiality of *RRM1* in 13 samples out of 21 (in four samples of JJN-3, six samples of H929, the unique sample of KMS-28-BM, and two samples of U266).

The same analysis was repeated for all the gMCSs (Supplementary Figs. 3–22). Note that not all gMCSs are equally important and they do not present the same explanatory power regarding the essentiality of *RRM1* in MM. The importance of each gMCS was measured by means of a Binomial test, where the statistical significance of the frequency of appearance of each gMCS within the samples was assessed (see "Methods" section). After correcting the *p*-values, six gMCSs remain significant, namely, $gMCS_2$, $gMCS_3$, $gMCS_4$, $gMCS_6$, $gMCS_9$ and $gMCS_{11}$ (Supplementary Table 5). Considering these six gMCSs, we could explain the essentiality of *RRM1* in all samples considered, except for two samples of U266, which questions the essentiality of *RRM1* in this cell line.

In order to provide a more objective comparison with experimental data, we summarized the results obtained at the cell line level (see "Methods" section). JJN-3, H929 and KMS-28-BM resulted to have at least one gMCS with all partner genes unexpressed while U266 did not. As a consequence, our gMCS approach concluded that *RRM1* is essential in JJN-3, H929, and KMS-28-BM, but not in U266.

We also conducted the essentiality analysis of *RRM1* on reconstructed networks of the four cell lines considered using GIMME and iMAT. GIMME predicted the essentiality of *RRM1* in all cell lines except for H929, while iMAT identified *RRM1* as essential in JJN-3 and H929, but not in KMS-28-BM and U266. See Supplementary Table 6 for further details.

We carried out an experimental validation of the aforementioned hypothesis. We examined the effect of two different siRNAs specific to *RRM1* on cell proliferation of four MM cell lines (JJN-3, H929, KMS-28-BM and U266). Both *RRM1* siRNAs efficiently decreased *RRM1* expression in the four cell lines analyzed as detected by qRT-PCR (Fig. 3a). Downregulation of *RRM1* expression with any of the two siRNAs significantly reduced cell proliferation (Fig. 3b) in the first three MM cell lines tested, but not in U266. Note that we also observed apoptosis

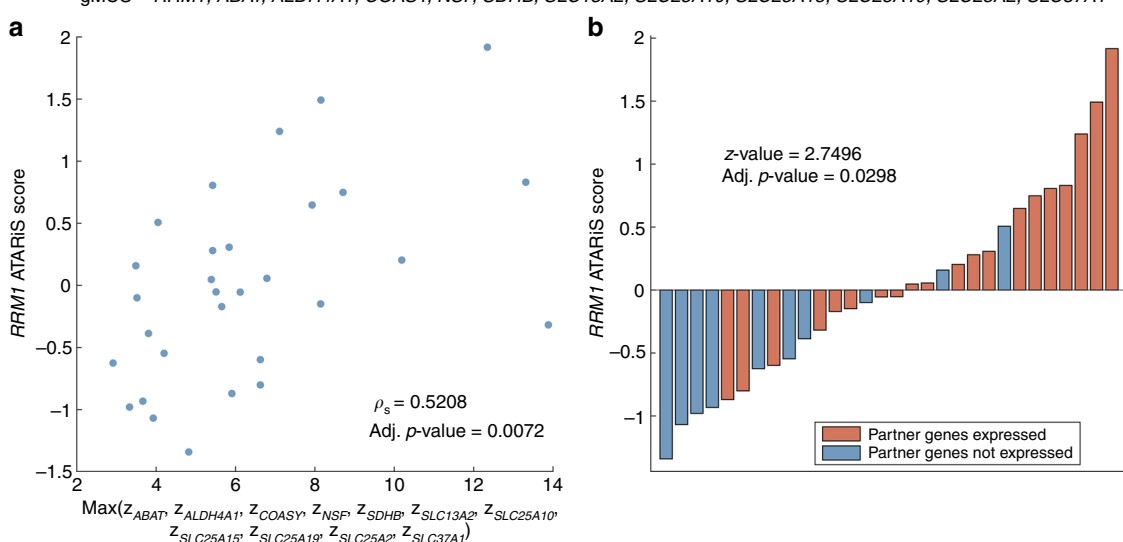

**Fig. 4** Sensitivity to *RRM1* silencing in the top 30 Achilles cell lines regarding quality and expression of partner genes in gMCS$_6$. **a** Spearman's correlation analyzing the dependence of *RRM1* ATARiS score on gene expression levels of partner genes in gMCS$_6$. **b** Bar plot of *RRM1* ATARiS score when partner genes are (or not) expressed in gMCS$_6$. To decide whether a gene is expressed, we used the standard threshold provided by the Gene Expression Barcode algorithm 3.0[23]

induction in JJN-3, H929, and KMS-28-BM after *RRM1* expression downregulation (Supplementary Fig. 23). These results support the essential role of *RRM1* in JJN-3, H929, and KMS-28-BM but not in U266, as predicted by our algorithm. Regarding GIMME and iMAT, the essential role of *RRM1* is not well predicted in two and one cell lines, respectively. These results show again that our approach outperforms competing approaches in the literature. Importantly, the role of *RRM1* in MM seems to depend on the transcriptomic profile of cell lines considered.

*Synthetic lethality and RRM1:* in contrast with reconstruction-based methods (such as iMAT or GIMME), our approach not only allows us to predict metabolic targets but to hypothesize the mechanism under which the knockout strategy is effective. Particularly for the *RRM1* knockout, given that identified gMCSs are synthetic lethals, we expect a higher essentiality of *RRM1* when we have a lower expression of its partner genes in the gMCSs identified. This could lead to disentangle the distinct outcome of U266 with respect to JJN-3, H929 and KMS-28-BM after *RRM1* inhibition.

To provide a general validation of this hypothesis regarding the aforementioned gMCS, again we used genome-scale loss-of-function screens provided by the Project Achilles v2.4.3[19]. In this case, however, we used the ATARiS score presented in Shao et al.[29] as a measure of essentiality for the gene under analysis, in our case, *RRM1*. Note that the ATARiS score ranks cell lines according to their level of sensitivity to the silencing of a particular gene and, therefore, it is suitable for the analysis proposed here. This data was downloaded from the Project Achilles Data Portal. Finally, the gene expression data was normalized with the Gene Expression Barcode algorithm 3.0[23] using standard parameters.

Figure 4a shows the relationship between the ATARiS score of *RRM1* in the 30 cell lines included in the Achilles Project and the expression of the *RRM1* partner genes in the gMCS discussed in Fig. 2, i.e., gMCS$_6$. Note that we took the maximum expression level among the different partner genes for each cell line (Supplementary Data 1). This is because each gene represents an escape pathway for the silencing of *RRM1* and, therefore, the one with the highest expression restricts the most its effect.

If we use the average or sum of the expression of the partner genes of *RRM1*, we obtain similar results (Supplementary Table 7). Calculating Spearman's correlation for this case, we can observe that the trend obtained for the gMCS under study is the one expected and is indeed significant ($\rho_s$: 0.5208; adj. *p*-value: 0.0072).

Furthermore, in Fig. 4b *red bars* represent cell lines where at least one partner gene of the gMCS is expressed and *blue bars* show cell lines where all partner genes of the gMCS are lowly expressed. The height of the bars is the ATARiS score of *RRM1* in each cell line. We expect a higher ATARiS score for *RRM1* (i.e., a lower essentiality) in those cell lines where at least one partner gene of the MCS is expressed (*red bars*). In the same way as it happens with Fig. 4a, the result for the gMCS under study makes sense under our hypothesis. In addition, in order to give statistical significance to this analysis, we carried out a one-tailed Mann–Whitney test as done in Shao et al.[29]. In this case, those cell lines with expressed partner genes have a significantly higher ATARiS score for *RRM1* (adj. *p*-value = 0.0298).

The same analysis was conducted for all gMCSs identified in the previous section (Supplementary Figs. 3–22) and, overall, statistical significance analyses seem consistent (Supplementary Table 5). After carrying out a correction of *p*-values for multiple testing, six gMCSs remained significant regarding the scatterplot (gMCS$_5$, gMCS$_6$, gMCS$_8$, gMCS$_{10}$, gMCS$_{18}$ and gMCS$_{20}$). These six gMCSs show a common pattern and differences mainly arise from the subunit introduced to delete the mitochondrial complex ATP synthase, which is known to be repressed in many tumor cells[30]. This result is certainly interesting because the dependence of *RRM1* in MM seems to be linked to the Warburg effect[31].

However, some gMCSs significant in the Binomial test do not exhibit such a behavior here. Using both Barcode and RNA-seq data (Supplementary Figs. 24 and 25), we could provide further insights for this result. For example, gMCS$_2$–gMCS$_4$ involve *GUK1*, which is highly expressed in MM, according to RNA-seq data, which restricts the explanatory power of these three gMCSs. This is not an issue since gMCS$_6$ (individually), the only significant gMCS in all statistical analyses developed in this

work, would be sufficient to explain the essentiality of *RRM1* in those MM cell lines where it has been found to be essential.

With the results presented above, we hypothesize that the essentiality of *RRM1* in MM (in light of the cell lines analyzed) depends on the expression of its partner genes in gMCS$_6$ (Fig. 2). This mechanism could be important in other tumors. To illustrate this, we carried out additional experimental validation in H23, a lung cancer cell line used in Fig. 4 where our algorithm predicted its dependence on *RRM1* using gMCS$_6$. Experimental results again confirmed our prediction (Supplementary Fig. 26).

## Discussion

This work presents, to the best of our knowledge, the first application of the concept of MCSs to cancer research. Thanks to the significant advances in the computation of MCSs made in the last years, we can more accurately and efficiently explore the solution space of knockout strategies to disrupt growth, going beyond exhaustive strategies typically limited to lower order synthetic lethals.

In this article, we adapted the computational framework presented in Tobalina et al.[15] in order to calculate gMCSs (a generalization at the gene level of MCSs). gMCSs is a more natural concept of synthetic lethality, which accounts for non-trivial GPR rules, neglected when we focused on reaction knockouts. The integration of gene expression data allows us to exploit the concept of synthetic lethality for the identification of metabolic vulnerabilities in cancer. Although it is not considered in this work, the integration of other genomic events, such as mutations or copy number variations, is possible and will be considered in the future.

It is also important to emphasize that the search of gMCSs is conducted on the reference (uncontextualized) human genome-scale metabolic network, in our case Recon2.v04. This element of our approach presents two advantages: (1) avoiding unnecessary heuristic choices as done in context-specific reconstruction algorithms and (2) identification of possible testable mechanisms by which the inhibition of a certain target is lethal, information which is lost when using cancer-specific metabolic reconstructions. These features make it more accurate and informative than previous approaches in the literature, as shown by our side-by-side comparison with iMAT and GIMME.

The predictive power of our approach was illustrated in the study of *RRM1* inhibition in four different MM cell lines. We confirmed the essentiality of *RRM1* in JJN-3, H929, and KMS-28-BM, but not in U266, which illustrates the heterogeneity found in MM and the need of the approaches as the one shown here. In vitro experimental validation showed the relevance of *RRM1* to sustain growth in the same MM cell lines as predicted by our algorithm.

Our approach enables to spot potential mechanisms explaining the effectiveness of a particular gene knockout to disrupt growth. Again, this was illustrated in the study of *RRM1*, where, using genome-scale loss-of-function screens from Project Achilles[19], we observed that cell lines with a lower expression of the partner genes of *RRM1* in the gMCS$_6$ (reported in Fig. 2) are more sensitive to the anti-proliferative effect of *RRM1* silencing (Fig. 4). This result is important and shows the value of our gMCS approach presented here in comparison with existing methods, which eliminate this information when the reference metabolic network is contextualized. In our opinion, this information is certainly relevant and could open new avenues for the development of tools for personalized medicine in cancer, since the expression of partner genes of *RRM1* can be used as a signature to predict the response to *RRM1* inhibition.

Overall, our approach lays the foundation to build a novel family of algorithms to understand and target metabolism in cancer. We would like to clarify that our methodology is general and its success depends on the quality of the metabolic network, GPR rules, the definition of the biomass equation and a correct classification of genes into expressed/unexpressed. As this information is continuously improving day-by-day, we expect that our approach will gain in acceptance and general use in the area of personalized medicine in the following years.

## Methods

**gMCSs computation**. A metabolic network of $m$ metabolites and $n$ reactions is usually represented as an $m \times n$ stoichiometry matrix where reactions are organized into columns and metabolites into rows. For each reaction, positive coefficients correspond to products while negative ones to educts. The activity of the reactions is represented by the flux vector $\mathbf{r}$. Here, we split reversible reactions into two irreversible steps and, therefore, reaction fluxes are non-negative (Eq. (1)).

$$\mathbf{r} \geq 0. \tag{1}$$

Under the steady-state assumption (represented in Eq. (2)), the sum of fluxes which produce a certain metabolite must be equal to the sum of fluxes which consume it.

$$\mathbf{S} \cdot \mathbf{r} = 0. \tag{2}$$

To calculate $\mathbf{S}$ in the analyses conducted in "Results" section, we applied fastFVA[32] to Recon2.v04[22] under RPMI1640 growth medium conditions[6] and removed blocked reactions and directionalities.

Our goal is to block a given metabolic task employing the least number of gene knockouts. The metabolic task to target is shown in Eq. (3):

$$\mathbf{t}^{\mathrm{T}} \cdot \mathbf{r} \geq r^{*}, \tag{3}$$

being $\mathbf{t}$ a null vector with a 1 in the position of the reactions involved in the metabolic task to target and $r^*$ a positive constant. The target metabolic task in our case is the biomass reaction. We used the biomass reaction available in Recon2.v04.

We are interested in the minimum number of gene knockouts which would make the biomass reaction impossible to occur. In order to exploit the concept of synthetic lethality and identify metabolic vulnerabilities in cancer, we are particularly interested in knockouts from the set of lowly expressed genes ($L$). To that end, we introduce the binary $g \times n$ matrix $\mathbf{G}$, which defines for each row the set of blocked reactions arising from the knockout of a particular subset of genes in $L$. Genes associated with each row in $\mathbf{G}$ are functionally interrelated and their simultaneous knockout is required to delete at least one of the reactions in the metabolic network. The number of genes for each row in $\mathbf{G}$ is stored in the $g \times 1$ vector $\mathbf{a}$. Note that if we focus on reaction knockouts instead of gene knockouts, as in previous works[14, 15], in the case that all reactions are irreversible, $\mathbf{G}$ would be the identity matrix $\mathbf{I}_n$. In order to calculate $\mathbf{G}$, we used the GPR rules available in Recon2.v04. The use of $\mathbf{G}$ allows us to move from MCSs at the reaction level to MCSs at the gene level (gMCSs). For specific examples of $\mathbf{G}$ matrix and details of its calculation in our MM study, see Supplementary Note 4.

Based on the above, we include all the possible reaction knockouts arising from the combination of genes in $L$:

$$\mathbf{G} \cdot \mathbf{r} \leq 0. \tag{4}$$

Note here that we only limit the right-hand side in Eq. (4) because the irreversibility constraint represented in Eq. (1) limits the left-hand side.

Next, similarly to von Kamp and Klamt[14], we computed the dual problem (Eqs (5–7)) of the set of equations shown above.

$$\mathbf{N} \cdot \begin{pmatrix} \mathbf{u} \\ \mathbf{v} \\ w \end{pmatrix} = \begin{bmatrix} \mathbf{S}^{\mathrm{T}} & \mathbf{G}^{\mathrm{T}} & -\mathbf{t} \end{bmatrix} \cdot \begin{pmatrix} \mathbf{u} \\ \mathbf{v} \\ w \end{pmatrix} \geq \mathbf{0} \tag{5}$$

$$\mathbf{v} \geq \mathbf{0}, \, w \geq 0 \tag{6}$$

$$\mathbf{u} \in R^m, \, \mathbf{v} \in R^g, \, w \in R. \tag{7}$$

In order to simplify the understanding of this new set of equations, the dual problem can be viewed as a new stoichiometry matrix ($\mathbf{N}$) with a new set of flux variables, namely, $\mathbf{u}$, $\mathbf{v}$ and $w$. Following the duality theory, each variable of the dual problem is related to a constraint in the primal problem in the following way: the primal constraints related to dual variables which take values different from zero are active and limiting. In particular, $u_i$ variables stand for the steady-state constraints, $v_i$ variables refer to the flux limitation constraints associated to combinations of gene knockouts in $L$, and $w$ is the dual variable related to Eq. (3). As explained in Ballerstein et al.[13], because the primal problem is infeasible, the dual problem is unbounded. Importantly, elementary flux modes (EFMs) of

the dual problem which contain $w$ and have minimal support in $\mathbf{v}$ will be gMCSs of the primal problem. In order to force $w$ to be active, we included the following constraint:

$$-r^* \cdot w \leq -c, \; c > 0, \tag{8}$$

where $c$ is a positive constant that forces Eq. (3) (in the primal problem) to be a limiting constraint.

So far, we are able to compute gMCSs which deprive the metabolic network from carrying out a given task by calculating the EFMs of its dual network. In order to calculate gMCSs which pass through a particular gene knockout, however, we need to include additional constraints[15]. Firstly, we force the gene knockout of interest to be included in the solution:

$$\begin{pmatrix} \mathbf{0} & \mathbf{d} & 0 \end{pmatrix} \cdot \begin{pmatrix} \mathbf{u} \\ \mathbf{v} \\ w \end{pmatrix} \geq b, \; b > 0, \tag{9}$$

where $\mathbf{d}$ is a null 1 x $g$ vector with a "1" in the position of vector $\mathbf{v}$ related with the gene knockout of interest. For example, in our MM study, we searched for gMCSs involving the knockout of RRM1. Note that, because of its knockout, we included RRM1 in the set $L$.

Secondly, in order to guarantee that the solution is still a gMCS, we need to ensure that Eq. (9) is redundant with respect Eqs (5–8), as detailed in Tobalina et al.[15]. To that end, we included Eqs (10–12), which force Eq. (9) to be written as a linear combination with non-negative coefficients ($x_i$ variables) of Eqs (5 and 8). Such linear combination constraint only applies to active dual variables. For this reason, $\varepsilon_i$ and $\delta_i$ variables are introduced, being both non-negative.

$$\begin{bmatrix} \mathbf{S} & \mathbf{0} \\ \mathbf{G} & \mathbf{0} \\ -\mathbf{t}^\mathrm{T} & r^* \end{bmatrix} \cdot \mathbf{x} = \begin{pmatrix} \mathbf{0} \\ \mathbf{d} + \varepsilon - \delta \\ 0 \end{pmatrix} \tag{10}$$

$$\varepsilon \geq \mathbf{0}, \; \delta \geq \mathbf{0}, \; \mathbf{x} \geq \mathbf{0} \tag{11}$$

$$\varepsilon \in R^g, \; \delta \in R^g, \; \mathbf{x} \in R^{n+1}. \tag{12}$$

Equations (13–15) allow us to link binary $z_i$ variables with $v_i$ variables ($z_i = 0 \leftrightarrow v_i = 0, z_i = 1 \leftrightarrow v_i > 0$) and $\varepsilon_i$ and $\delta_i$ ($z_i = 1 \rightarrow \varepsilon_i = \delta_i = 0$):

$$\alpha \cdot \mathbf{z} \leq \mathbf{v} \leq M \cdot \mathbf{z} \tag{13}$$

$$M \cdot (\mathbf{1} - \mathbf{z}) \geq \varepsilon + \delta \tag{14}$$

$$\mathbf{z} = [z_1, z_2, \dots, z_g], \; z_i \in \{0, 1\}, \tag{15}$$

where $\alpha$ and $M$ are small and large positive constants, respectively.

The set of constraints presented above guarantees that any feasible solution (if any exists) includes at least one gMCS involving the gene knockout of interest. In order to obtain exactly one gMCS, we need to minimize the number of gene knockouts involved.

The objective function used here was the following:

$$\text{minimize} \sum_{i=1}^{i=g} a_i \cdot z_i, \tag{16}$$

where $a_i$ represents the number of genes in $L$ associated with a given dual $v_i$ variable, which is equivalent to find the gMCS that minimizes the total number of genes in $L$.

It is important to emphasize that, if optimality is achieved, the solution is certainly a gMCS. The reason comes from the constraints presented above, which guarantee that any feasible solution does include at least one gMCS involving the gene knockout of interest. Therefore, by minimizing the number of genes in $L$, a gMCS is obtained. If optimality is not achieved, we can only guarantee that the solution returned includes one gMCS involving the gene knockout of interest, but it will not necessarily be the optimal one.

Finally, we can iteratively enumerate gMCSs by introducing a new constraint that eliminates previously obtained solutions, as done in previous works[14, 33]:

$$\sum_{i=1}^{i=g} z_i^k z_i \leq \sum_{i=1}^{i=g} z_i^k - 1. \tag{17}$$

For the different studies conducted in this article, we fixed a five-minutes time limit to find a gMCS on a 64 bit Intel Xeon E5-2670 at 2.60 GHz (16 cores) and 64 GB of RAM. MATLAB was used to implement the algorithm, with help of IBM Ilog Cplex for the underlying mixed-integer linear programming model. The most time consuming step was the calculation of the $\mathbf{G}$ matrix, requiring ~20 min.

**Gene and reaction classification**. In order to build $\mathbf{G}$ and $\mathbf{a}$ matrices, which are required in Eqs. (5, 10 and 16), respectively, we need to determine the set of lowly expressed genes and reactions. This was done from gene expression experiments, in our case collected from GEO database[34]. We focused on Affymetrix HGU133plus2 arrays, which can be processed using Barcode[23]. This method is designed to be able to work with just one sample and make it comparable to others, instead of needing several samples at the same time. We preprocessed the data using Barcode's R script, using one sample at a time. We retrieved the $z$-score obtained from this algorithm, which is equivalent to processing each sample with fRMA[35].

Because the $z$-scores retrieved from Barcode were given at the probe-set level, using gene-probe relationships annotated in hgu133plus2.db R package, we obtained the gene $z$-score value as the median value of the corresponding $z$-scores of its associated probe-sets. In case we have several samples for the same condition and we want to obtain a consensus expression (e.g., at the cell line level), we took the median value of the expression of each gene in the samples considered. Each gene value was transformed into present(1)/absent(0) call using Barcode's criteria, i.e., genes with $z$-score $\geq 5$ are expressed. Absent genes are stored in $L$.

Similarly, reactions are classified into $L$ set using gene–protein-reaction rules annotated in Recon2.v04 and the gene expression classification described above. These rules establish a relationship between the enzymes that catalyze each reaction and the genes that code for those enzymes[36]. A reaction may be catalyzed by a single enzyme, different isozymes or a protein complex. Reactions having OR rules associated can be catalyzed by different isozymes, while those having AND rules involve protein complexes.

If a reaction is associated with a single gene, it is classified as $L$ if its corresponding gene is also in $L$. If it involves an OR rule, it is classified as $L$ if all the genes are classified as $L$. If a reaction involves an AND rule, it is classified as $L$ if any of the genes is classified as $L$.

**Cell culture**. The MM cell lines KMS-28-BM, H929, U266, and H23 were maintained in culture in RPMI1640 medium supplemented with 10% fetal bovine serum (Gibco, Grand Island, NY), penicillin/streptomycin (BioWhitaker, Walkersvill, MD) at 37 °C in a humid atmosphere containing 5% $CO_2$. JJN-3 were cultured with 40% DMEM, 40% IMDM, and 20% fetal bovine serum. Cell lines were obtained from the DSMZ or the American Type Culture Collection (ATCC). All cell lines were authenticated by performing an short tandem repeat allele profile and were tested for mycoplasma (MycoAlert Sample Kit, Cambrex), obtaining no positive results.

**Cell transfection**. Cells were passaged 24 h before nucleofection, and cells for nucleofection were in their logarithmic growth phase. The transfection of siRNAs was done with the Nucleofector II device (Amaxa GmbH, Köln, Germany) following the Amaxa guidelines. Briefly, $1 \times 10^6$ of JJN-3, H929, KMS-28-BM, and U266 cells were resuspended in 100 µL of supplemented culture medium with 50 nM of RRM1 siRNAs or Silencer Select Negative Control-1 siRNA (Ambion, Austin, TX) and nucleofected with the Amaxa nucleofector apparatus using programs G-016, A-033, A-023 and X-005, respectively. In the case of H23, siRNAs were transfected using lipofectamine transfection reagent 2000 (Invitrogen, Carlsbad, CA) according to manufacturer's protocol. Briefly, H23 cells (20,0000 cells per well) were seeded in a six-well plate with antibiotic-free medium 24 h before transfection. Cells were then incubated with transfection mixtures containing 50 nM of siRNAs or Silencer Select Negative Control-1 siRNA (Ambion, Austin, TX) for 4 h. Then, medium was replaced with full culture medium. We used two different siRNAs against RRM1 target (siRRM1 A: GGAUCGCUGUCUCUACUU; siRRM1 B: AGAUCUUUGAAACUAUUUA) to demonstrate that the results obtained with RRM1 siRNA nucleofection are not due to a combination of inconsistent silencing and sequence specific off-target effects. Silencer Select Negative Control-1 siRNA was used to demonstrate that the nucleofection did not induce non-specific effects on gene expression. Nucleofection was performed twice with a 24 h interval. After 48 h of the second nucleofection, the RRM1 mRNA expression was analyzed by qRT-PCR (GUS was employed as the reference gene). Cell proliferation was analyzed 0, 2, 4 and 6 days after two repetitive transfections. Transfection efficiency was determined by flow cytometry using the BLOCK IT Fluorescent Oligo (Invitrogen Life Technologies, Paisley, UK).

**Cell proliferation assay**. Cell proliferation was analyzed using the CellTiter 96 Aqueous One Solution Cell Proliferation Assay (Promega, Madison, W). This is a colorimetric method for determining the number of viable cells in proliferation. For the assay, 100 µL of nucleofected cells were plated in 96 wells plates 0, 2, 4 and 6 days after the last nucleofection. Plates with suspension cells were centrifuged at $800 \times g$ for 10 min and medium was removed. Then, cells were incubated with 100 µL per well of medium and 20 µL per well of CellTiter 96 Aqueous One Solution reagent. After 1–3 h of incubation at 37 °C, the plates were incubated for 1–4 h, depending on the cell line at 37 °C in a humidied, 5% $CO_2$ atmosphere. The absorbance was recorded at 490 nm using 96-well plate readers until absorbance of control cells without treatment was around 0.8. The background absorbance was measured in wells with only cell line medium and solution reagent. First, the average of the absorbance from the control wells was subtracted from all other absorbance values. Data were calculated as the percentage of total absorbance of RRM1 transfected cells/absorbance of control cells.

**Apoptosis assay**. The FITC Annexin V Apoptosis Detection Kit I (cat. no. 556419, BD Pharmingen) was used following the manufacturer's instructions, with some modification. Firstly, 100 μL of nucleofected cells were washed twice with phosphate-buffered saline and resuspended in 1× binding buffer at a concentration of $1 \times 10^6$ cells per mL. 1 μL of FITC Annexin V (AV) antibody and 2 μL of propidium iodide (PI) were added and incubated for 15 min at RT in the dark. Finally, 400 μL of 1× binding buffer were added to each tube and analyzed by flow cytometry within 1 h. We represented the addition of FITC AV positive and PI negative cells (early apoptosis) and FITC AV positive and PI positive cells (end stage apoptosis, death).

**Quantitative RT-PCR**. The expression of *RRM1* was analyzed by qRT-PCR in JJN-3, H929, KMS-28-BM MM and U266 cell lines. First, total mRNA was extracted with Trizol Reagent 5791 (Life Technologies, Carlsbad, CA, USA) following the manufacturer instructions. RNA concentration was quantified using NanoDrop Specthophotometer (NanoDrop Technologies, USA). cDNA was synthesized from 1 μg of total RNA using the PrimeScript RT reagent kit (Perfect Real Time) (cat. no. RR037A, TaKaRa) following the manufacturer's instructions. The quality of cDNA was checked by a multiplex PCR that amplifies *PBGD, ABL, BCR*, and *β2-MG* genes. qRT-PCR was performed in a 7300 Real-Time PCR System (Applied Biosystems), using 20 ng of cDNA in 2 μL, 1 μL of each primer at 5 μM (RRM1 F:5′-AAAGAGCAACCAGCAGAACC-3′; RRM1 R:5′CCAGGGAAGCCAAATTA CAA-3′; GUS F:5′GAAAATATGTGGTTGGAGAGCTCATT-3′; GUS R:5′-CCG AGTGAAGATCCCCTTTTTA-3′), 6 μL of SYBR Green PCR Master Mix 2X (cat. no. 4334973, Applied Biosystems) in 12 μL reaction volume. The following program conditions were applied for qRT-PCR running: 50 °C for 2 min, 95 °C for 60 s following by 45 cycles at 95 °C for 15 s and 60 °C for 60 s; melting program, one cycle at 95 °C for 15 s, 40 °C for 60 s and 95 °C for 15 s. The relative expression of each gene was quantified by the Log2($-\Delta\Delta$Ct) method using the gene *GUS* as an endogenous control.

**Statistical analyses**. In Results section "Genetic minimal cut sets and cancer-specific essential genes", we conducted a logistic regression to investigate whether our gMCS approach recovers as essential more Achilles-based essential cases than Achilles-based non-essential cases:

$$\log\left(\frac{p(\text{essential in Achilles})}{1-p(\text{essential in Achilles})}\Big|x\right) = \beta_0 + \beta_1 \cdot x, \tag{18}$$

where $x = \{0, 1\}$, being 1 when our gMCS approach returns an essential gene, 0 otherwise. We evaluated the statistical significance of $\beta_1$, whose magnitude can be transformed to the OR as follows:

$$\text{OR} = \exp(\beta_1) = \frac{\frac{p(\text{essential in Achilles}|x=1)}{1-p(\text{essential in Achilles}|x=1)}}{\frac{p(\text{essential in Achilles}|x=0)}{1-p(\text{essential in Achilles}|x=0)}} \tag{19}$$

In "Results" section "*RRM1* essentiality in MM", for each gMCS, we calculated an adjusted *p*-value using the one-sided Binomial test in order to evaluate whether its frequency in the MM samples considered is significantly high. In particular, for the null hypothesis, we used a conservative strategy and fixed $p = 0.25$, based on the analysis conducted in section "Genetic minimal cut sets and cancer-specific essential genes", where we could infer, using the Project Achilles data as gold-standard, the probability of obtaining a false positive with our approach, namely: *p(nonessential in Achilles | essential gene in gMCS approach)*.

Finally, in "Results" section "Synthetic lethality and *RRM1*", for each gMCS, the Spearman's correlation was used to calculate the linear association between the *RRM1* ATARiS score and the gene expression levels of the partner genes of *RRM1* of considered Project Achilles cell lines. In addition, for each gMCS, the Mann–Whitney test was used to compare the *RRM1* ATARiS score in two different groups: (1) considered Project Achilles cell lines where all partner genes of *RRM1* are expressed and (2) considered Project Achilles cell lines where at least one partner gene is expressed. Both statistical analyses have been carried out one-tailed.

In order to correct the *p*-values in different analyses considered above, we used the FDR approach available in the function *p.adjust* of R software.

**RNA sequencing data**. We used Illumina RNA-seq data generated in Agirre et al.[37], which involves 11 samples of MM patients and four samples of normal plasmatic cells from tonsils. Reads in raw FASTQ files were aligned to the human reference genome (hg19) using TopHat version 2.0.9[38]. Annotated reads were then assembled using Cufflinks 2.02[39] and Gencode v.19[40] as reference file. We used CuffDiff2 to estimate gene abundance in log2(fpkm).

**Code availability**. The code used to generate the results shown in this article is provided as Supplementary Software.

**Data availability**. The authors declare that all data supporting the findings of this study are available within the article and its Supplementary Materials, or from the corresponding authors upon request.

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

## Acknowledgements

I.A. was supported by a Basque Government predoctoral grant [PRE_2016_2_0044]. This work was supported by the Minister of Economy and Competitiveness of Spain [BIO2013-48933, BIO2016-77998-R], ELKARTEK Programme of the Basque Government [KK-2016/00026], Centro de Ingeniería Biomédica (University of Navarra), Instituto de Salud Carlos III (ISCIII) [PI10/01691], [PI13/01469], [PI14/01867], [PI16/02024], [RTICC RD12/0036/0068], CIBERONC [CB16/12/00489] (Co-finance with FEDER funds), ERA-NET programmes TRANSCAN-2 JTC EPICA by the "Torres Quevedo" Subprogramme [PTQ-11-04777], [PTQ-14-07320 I.D.M], Gobierno de Navarra [40/2016] and Fundació La Marató de TV3 [20132130-31-32]. We would like to thank Angel Rubio for his support in statistical analyses.

## Author contributions

X.A., F.P. and F.J.P. conceived this study. I.A., L.T. and F.J.P. developed the gMCS approach and I.A. carried out its computational implementation. E.S.J.-E., E.M., L.G. and X.A. performed the experiments. All authors wrote, read and approved the manuscript.

## Additional information

**Competing interests:** The authors declare no competing financial interests.

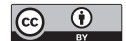

