## [Peer Review File · Nature Communications]

Reviewers' comments:

Reviewer #1 (Remarks to the Author):

In their work, Apaolaza et al. present an approach to identify genetic minimal cut sets (gMCSs) to prevent cancer proliferation. Using their approach they study the essentiality of a particular enzyme, RRM1, in a specific cancer type for which three cell lines are available. While the application of the concept of gMCSs to study cancer metabolism and its vulnerabilities is quite interesting my feeling is that the scope of the application is too limited to really demonstrate its usefulness.

Major points:

Throughout the discussion, the authors concentrate mostly on a single gCMS though they calculated 18 in total. It does not really become clear why this gCMS was chosen but it would be more sensitive to discuss all gCMs and not just those that one that probably fit the story line well. In that case, the authors should think about which type of summary statistics they provide in order to show that gCMSs indeed provide more information than what was known before.

While an experimental validation is definitively something that can help validating computational predictions, in the present case it doesn't really add to the manuscript. As the authors already acknowledge it has already been reported that RRM1 is a promising target in multiple myeloma (MM) and that's the reason why they chose that enzyme as the focus of their study. In consequence it is expected that MM cell lines show slower growth if that enzyme is silenced. Hence, the argument becomes circular as the author's approach does not return RRM1 as a promising target but rather they already chose it a priori. It would be more sensible to consider other targets just than the single one and show that promising targets beyond RRM1 are indeed more often reflected in gMCSs containing many lowly-expressed genes. As it stands, the experimental part does not really provide much novel information and should be toned much down since it is not a validation but just a confirmation of the basic a priori assumption that RRM1 is important in those cell lines as previously reported for MM in general.

While the analysis of the RRM1 over a large number of cancer cell lines (Achilles data) is a more robust way of validation there are some uncertainties regarding the statistics reported. The author mostly discuss a single gCMS and provide some p-values. However, the authors have a total of 18 gMCSs and it is not mentioned in the manuscript whether they have actually corrected the corresponding p-values for multiple testing (there was no mention of multiple test correction in the manuscript) and hence it is unclear whether any of these p-values is indeed significant after test correction. Again, it would be more sensitive to provide information across all gMCSs and not just a selected one and subsequently move to that gMCSs to probably discuss some details.

It is not clear whether the comparison to other approaches is really fair (see also my comment on the exclusion of iMAT in the list of minor points). The authors call GIMME less accurate since it predicts essentiality of RRM1 only in ten out of 18 of the considered data sets. First, this comparison is considerably inflated since basically only three cell lines are considered (some with several instances of expression data, some of them replicates from the same experiment). Thus, the test should rather be performed on the level of cell lines or at least expression data should be grouped according to experiment. Second, results from the gCMS approach are not really comparable to results returned from GIMME since they basically return different things – networks returned from GIMME allow to directly determine essentiality of RRM1 while based on the gCMS approach the authors check whether for each data set there exists at least one gMCSs that only contains lowly expressed genes besides RRM1. Thus, one would already by chance expect that gMCSs more likely reports lethality (since for each data set there are 18 gMCSs that are tested).

Minor points:

The coloring scheme in Fig. 2A is misleading. If 5 is the cut-off for an enzyme to be expressed, this

should be white. Otherwise it is quite difficult to grasp what particular shade of red actually indicates the cut-off for considering an enzyme as expressed.

In l. 241 the authors state that they did not consider iMAT due to its high computational demand and GIMME providing better results in a previous test of several methods for building context-specific metabolic networks. There is some criticism concerning the fairness of these previous tests (Machado & Herrgard, 2014) including that they have been performed on prokaryotic metabolic networks (which are focused on growth and the validation data considering fluxes in growing bacteria) while methods for reconstructing context-specific metabolic networks have typically been developed for eukaryotic networks (where rapid growth typically is not an ultimate imperative). Moreover, from personal experience I know that depending on the choice of cut-offs applying iMAT typically requires only seconds and rarely in some severe cases (depending on the choice of parameters) probably a couple of hours for determining context-specific networks in Recon v2.04. Thus, it should be possible to build these networks for the just 18 samples the authors are considering and include iMAT in the comparison.

In l. 265-270 the authors state that they used the gene with highest expression as the one most important for each gCMS. As there is no one-to-one relationship between abundance and importance of a gene, the authors should check whether they probably haven't always used the expression of the same gene as proxy for the expression strength of partner genes of RRM1. Beyond that it would also be more sensible to probably consider an aggregate of expression of partner genes of RRM1 and not just a single one.

The authors state that they used all gene expression data originating from a specific chip for determining lowly expressed genes. Since gene expression in cancer cell lines is probably strongly different from gene expression in normal human tissue or cell lines, they should check how robust their results are if considering, for instance, just gene expression from cancer cell lines (e.g. NCI-60).

Reviewer #2 (Remarks to the Author):

Summary

In the manuscript by Apaolaza et al., the authors present a method that uses the concept of gene minimum cut sets (gMCSs) to predict and exploit synthetic lethality in cancer metabolism. They first introduce a novel method to expand the concept of minimum cut sets from the reaction level to the gene level. Then, they apply this method to explore the effect of knock down of the RRM1 gene in 18 different gene expression contexts/instances involving three cell lines of multiple myeloma. Their method predicts RRM1 to be essential in 17 of these 18 samples. Next, they validate experimentally the essentiality of RRM1 using shRNA in the three cell lines experiments. Finally, they report a correlation between the expression of the gMCSs group in 30 different cell lines and the ATARiS essentiality score of RRM1 measured in shRNA screens.

We believe that this is innovative work with a potentially promising route to personalized medicine and, especially, for the much needed patient stratification application. Nevertheless, we feel that the method presented here requires considerable additional experimental validation to establish its predictive power before we can support its publication in Nature Communications.

Major Comments

1. Further experimental validation of the method. In this work, the authors present a method that can be used to predict the essentiality of a gene in a certain context. This can be easily tested using publically available shRNA screens for multiple gene knockdowns. To establish the general predictive power of their method, the authors should prove the ability of this method to predict the

essentiality of genes other than RRM1 in multiple cell lines using shRNA screens and the gene expression of these cell-lines. Only then, and if and only if, this approach may be of interest to the wide readership of Nature Communication. Note that this does not require new experiments! Just take a published collection of genome wide knockdowns of genes across many different cell-lines (e.g., like Achilles) and the associated cell-line transcriptomics, and overlay that expression data on the predicted gMCs of the different genes, and show that you can predict their essentiality!...

2. As the shRNA validation of the RRM1 gene essentiality is performed in cell lines and not samples, it would be preferred if the authors would predict essentiality in a cell line specific manner.

3. To establish the clinical relevance of this method, the authors could show that the expression level of the gMCS groups that they find is also relevant in in clinical data, e.g. in predicting patient survival in the TCGA cohort (its quite stragithforward to test).

Minor Comments

1. Please check the writing style so that methods are all described in the methods section and the results are listed subsequent order in the results section.

2. Please provide statistical estimates for the reported finding. E.g in line 201 to line 203, there should be a p-value representing he significance of these results.

Response to the Reviewer's comments (manuscript NCOMMS-16-24429-T)

We would like to thank the reviewers for their thorough revision, constructive criticisms and suggestions on our manuscript. We believe that this revised version of our study has clearly improved following the reviewer's advice.

Below you can find the detailed answers to all the issues raised by the distinguished reviewers. Note that the major changes introduced in the main text have been highlighted in red colour.

Reviewer #1 (Remarks to the Author):

In their work, Apaolaza *et al.* present an approach to identify genetic minimal cut sets (gMCSs) to prevent cancer proliferation. Using their approach they study the essentiality of a particular enzyme, RRM1, in a specific cancer type for which three cell lines are available. While the application of the concept of gMCSs to study cancer metabolism and its vulnerabilities is quite interesting my feeling is that the scope of the application is too limited to really demonstrate its usefulness.

Major points:

Throughout the discussion, the authors concentrate mostly on a single gMCS though they calculated 18 in total. It does not really become clear why this gMCS was chosen but it would be more sensitive to discuss all gMCSs and not just those that one that probably fit the story line well. In that case, the authors should think about which type of summary statistics they provide in order to show that gMCSs indeed provide more information than what was known before.

Response: We completely agree with the comment of the reviewer. In the section "RRM1 essentiality in Multiple Myeloma", we show one of the 18 gMCSs that better explains the essentiality of RRM1 in Multiple Myeloma (gMCS₆). A more general discussion is carried out in the section "Achilles data and RRM1". However, we recognize that this discussion is certainly incomplete and acknowledge the points made by both reviewers in order to improve and clarify this part of the manuscript. Note here that after improving the analysis in this new version of the manuscript (by considering one additional cell line), we obtained 20 gMCSs (see Results section, Page 8, 2nd paragraph)

First, in the section "RRM1 essentiality in Multiple Myeloma", we now provide a statistical significance analysis of identified gMCSs. In particular, for each gMCS, we calculated an adjusted p-value using the one-sided Binomial test, namely to evaluate whether its frequency in the MM samples considered is statistically significant. In particular, for the null hypothesis, we used a conservative strategy and fixed $p=0.25$, based on the analysis conducted in the previous subsection "Minimal Cut Sets at the gene level (gMCSs) and cancer-specific essential genes", where we could infer (using Achilles data as gold-standard) the probability of obtaining a false positive with our gMCS approach, namely: p (nonessential in Achilles| essential gene in gMCS approach) (see "Statistical Analyses" in Methods section). In order to correct the p-values, we used the FDR approach available in the function *p.adjust* of R software. As a result, 6 gMCSs remain significant (gMCS₂₋₄, gMCS₆, gMCS₉, gMCS₁₁, FDR $\leq 5\%$) (**Supplementary Table 5**, see below), among which the one shown in Figure 2 is included (gMCS₆). We included this analysis in the main text (1st paragraph, page 9). Note here that these gMCS explains the essentiality of RRM1 in MM samples significantly more often than expected by chance, according to the null hypothesis.

Secondly, as suggested by the reviewer below, we corrected the p-values of identified gMCSs for multiple hypothesis testing in the section "Achilles data and RRM1" (now "Synthetic lethality and RRM1"). Again, we used the FDR approach available in the function *p.adjust* of R software. In this case, 5 gMCSs remain significant after the correction (gMCS₅, gMCS₆, gMCS₈, gMCS₁₀, gMCS₁₈, FDR $\leq 5\%$) (**Supplementary Table 5**), among which the one shown in Figure 2 is included (gMCS₆). These results have also been added in the main text (Results section, page 13, 2nd paragraph).

Taking into account these two independent analyses (**Supplementary Table 5**), we have rephrased the discussion previously included in the main text (4th paragraph, page 13). Interestingly, the only gMCS being statistically significant in both analyses is gMCS₆, precisely the one previously shown in Figure 2 and Figure 4. In light of the cell lines analyzed, our main conclusion is that the essentiality of RRM1 in MM and possibly in cancer (we add a similar study in a single cell line in lung cancer) is dependent on the expression of its partner genes in gMCS₆. In our view this conclusion reveals an important new insight about RRM1 and cancer.

Supplementary Table 5: Summary of adjusted p-values for each gMCS found in the RRM1 essentiality study in Multiple Myeloma (MM) for different statistical analyses conducted in the main text.

	Binomial Test		Achilles Scatter Plot		Achilles Bar Plot	
	p-value	adj. p-value	p-value	adj. p-value	p-value	adj. p-value
gMCS₁	0.9255	0.9810	0.7422	0.7624	0.7892	0.9285
gMCS₂	0.00001	0.0007	0.7624	0.7624	0.3587	0.4782
gMCS₃	0.00001	0.0007	0.7009	0.7624	0.3587	0.4782
gMCS₄	0.00001	0.0007	0.7009	0.7624	0.3587	0.4782
gMCS₅	0.4334	0.9631	0.0012	0.006	0.0062	0.031
gMCS₆	0.0004	0.0019	0.0018	0.0071	0.003	0.0298
gMCS₇	0.0206	0.0589	0.0953	0.1466	0.0922	0.2049
gMCS₈	0.0561	0.1404	0.0037	0.0122	0.0207	0.0694
gMCS₉	0.0064	0.0214	0.064	0.1067	0.1542	0.3085
gMCS₁₀	0.9255	0.9810	0.0004	0.0047	0.0208	0.0694
gMCS₁₁	0.0064	0.0214	0.1501	0.2001	0.3848	0.4810
gMCS₁₂	0.9810	0.9810	0.0318	0.0794	-	-
gMCS₁₃	0.9810	0.9810	0.0318	0.0794	-	-
gMCS₁₄	0.8083	0.9810	0.1919	0.2398	0.0375	0.1071
gMCS₁₅	0.9255	0.9810	0.0534	0.097	0.2521	0.4202
gMCS₁₆	0.9255	0.9810	0.117	0.1671	0.0529	0.1322
gMCS₁₇	0.9255	0.9810	0.0534	0.097	0.2521	0.4202
gMCS₁₈	0.9255	0.9810	0.0006	0.0047	0.0056	0.031
gMCS₁₉	0.9255	0.9810	0.0472	0.097	-	-
gMCS₂₀	0.6326	0.9810	0.0007	0.0047	0.0008	0.0154

* In these cases, all cell lines were assigned to a single class and, therefore, the Mann-Whitney test could not be calculated.

While an experimental validation is definitively something that can help validating computational predictions, in the present case it doesn't really add to the manuscript. As the authors already acknowledge it has already been reported that RRM1 is a promising target in multiple myeloma (MM) and that's the reason why they chose that enzyme as the focus of their study. In consequence it is expected that MM cell lines show slower growth if that enzyme is silenced. Hence, the argument becomes circular as the author's approach does not return RRM1 as a promising target but rather they already chose it a priori. It would be more sensible to consider other targets just than the single one and show that promising targets beyond RRM1 are indeed more often reflected in gMCSs containing many lowly-expressed genes. As it stands, the experimental part does not really provide much novel information and should be toned much down since it is not a validation but just a confirmation of the basic a priori assumption that RRM1 is important in those cell lines as previously reported for MM in general.

Response: We agree with the reviewer that RRM1 has been previously implicated in Multiple Myeloma and this was indeed a criterion we chose for selection. However, to address this rightful concern from the reviewer we have performed 2 additional sets of experiments, firstly by identifying myeloma cells lines in which our gMCS algorithm does not predict RRM1 to be essential and other tumor cell lines in which RRM1 has not been described as being essential and in which our algorithm predicts its essentiality. In both cases experimental data demonstrate the prediction.

In the first case, we identified a cell line in MM (U266) where our algorithm does not return RRM1 as essential. This was based on our analysis of gMCSs and gene expression data. More particularly, in all gMCSs considered, including gMCS₆, we found at least one partner gene of RRM1 that is expressed in U266 (in contrast with J2N-3, H929 and KMS-28-BM). We provided experimental validation of the non-essentiality of RRM1 in U266 (see Figure 3 of the revised version of the manuscript). Therefore, our algorithm is able to capture in which cases the inhibition of RRM1 is (or not) lethal in MM. An additional experimental validation in H23, a lung cancer cell line used in Figure 4, where our algorithm predicted its dependence on RRM1 using gMCS₆. Experimental results again confirmed our prediction (see Supplementary Figure 26 in the revised version of the manuscript).

On the other hand, as suggested by the reviewer, we analyze the performance of our algorithm in a broader context and its capacity to identify other targets. In particular, we used the genome scale loss-of-function screens provided by the Project Achilles v2.4.3 (ref. 1), focusing on the same set of cell lines as in the previous version of the manuscript, namely the 30 cell lines with the highest quality score calculated as done in Hart *et al.*² and with expression data available in the Cancer Cell Line Encyclopedia³ (Supplementary Table 1). For each cell line, we derived the top-20 most and least essential metabolic genes, according to the essentiality score calculated in Hart *et al.*², which is more appropriate than the ATARiS score to rank genes within a particular cell line⁴. Overall, based on Achilles data, we potentially have 600 essentiality cases and 600 non-essentiality cases. This information was used as gold-standard and, therefore, was compared with the results obtained with our algorithm. Note that we fixed a time limit of 5 minutes for each of the 1200 cases considered (as in the previous version of the manuscript).

Within the time limit, we achieved a solution in 308 out of 600 Achilles-based essentiality cases and in 451 out of 600 Achilles-based non-essentiality cases. Out of 308 solutions, 167 were correctly identified as essential, while 141 incorrectly identified as non-essential. On the other hand, out of 451 solutions, 340 were correctly identified as non-essential and 111 incorrectly identified as essential. The results are summarized in the contingency table below (see Figure 1d now):

		gMCS approach		
		Essential	Non-Essential	Total
Achilles	Essential	167	141	308
	Non-Essential	111	340	451
	Total	278	481	759

Figure 1d: Contingency table with the essentiality predictions of the gMCS approach in the Project Achilles data.

We conducted a logistic regression to investigate whether our gMCS approach (significantly) recovers as essential more Achilles-based essential genes than Achilles-based non-essential genes:

$$\log\left(\frac{p(\text{essential gene in Achilles})}{1 - p(\text{essential gene in Achilles})} \mid x\right) = \beta_0 + \beta_1 \cdot x$$

, where $x=\{0,1\}$, being 1 when our gMCS approach returns an essential gene, 0 otherwise. We evaluated the statistical significance of β_1 , whose magnitude can be transformed to the Odds Ratio (OR) as follows:

$$OR = \exp(\beta_1) = \frac{\frac{p(\text{essential in Achilles} \mid x = 1)}{1 - p(\text{essential in Achilles} \mid x = 1)}}{\frac{p(\text{essential in Achilles} \mid x = 0)}{1 - p(\text{essential in Achilles} \mid x = 0)}}$$

The difference is highly significant ($p\text{-value}=3.78 \cdot 10^{-16}$) with an odds ratio (OR) of 3.62 (see Methods section). These results prove that promising targets are indeed more often reflected in gMCSs containing many lowly-expressed genes (in fact, gMCSs where all genes are lowly expressed except one gene), as required by the reviewer. Note also that our analysis revealed a large list of potential essential genes beyond RRM1 (Supplementary Data 1). We are indeed in debt with the reviewer for this question, since these results substantially improve our work and the scope of our algorithm.

Note also that the same analysis was conducted for GiMME⁵ and iMAT⁶ (Supplementary Table 2-3). GiMME obtained a significant result ($p\text{-value}=0.003$, $OR= 2.13$) but not iMAT ($p\text{-value}=0.44$, $OR= 1.22$). Our approach substantially improved GiMME and iMAT, being particularly relevant our gain in sensitivity: we recovered 167 Achilles-based essential metabolic genes out of 308 solutions (in a 5 minutes time limit), while GiMME and iMAT, out of 600 cases, only 49 and 35 successes, respectively. This result is certainly significant, as we partially overcome the issue of lack of sensitivity observed in previous approaches⁷. In addition, we keep a similar precision value than GiMME and iMAT (0.6-0.7).

Supplementary Table 2. Contingency table with the essentiality predictions of GiMME in the Project Achilles data.			
		GiMME	
		Essential	Non-Essential
Achilles	Essential	49	551
	Non-Essential	24	576

Supplementary Table 3: Contingency table with the essentiality predictions of iMAT in the Project Achilles data.			
		iMAT	
		Essential	Non-Essential
Achilles	Essential	35	565
	Non-Essential	29	571

These results have been summarized and included in the main text, Results subsection “Minimal Cut Sets at the gene level (gMCSs) and cancer-specific essential genes”.

While the analysis of the RRM1 over a large number of cancer cell lines (Achilles data) is a more robust way of validation there are some uncertainties regarding the statistics reported. The author mostly discuss a single gCMS and provide some p-values. However, the authors have a total of 18 gCMSs and it is not mentioned in the manuscript whether they have actually corrected the corresponding p-values for multiple testing (there was no mention of multiple test correction in the manuscript) and hence it is unclear whether any of these p-values is indeed significant after test correction. Again, it would be more sensitive to provide information across all gCMSs and not just a selected one and subsequently move to that gCMSs to probably discuss some details.

Response: As suggested by the reviewer, we have included these statistics in a new version of the manuscript (**Supplementary Table 5**). This question has also been directly addressed in the answer to the first question of the reviewer.

It is not clear whether the comparison to other approaches is really fair (see also my comment on the exclusion of iMAT in the list of minor points). The authors call GIMME less accurate since it predicts essentiality of RRM1 only in ten out of 18 of the considered data sets. First, this comparison is considerably inflated since basically only three cell lines are considered (some with several instances of expression data, some of them replicates from the same experiment). Thus, the test should rather be performed on the level of cell lines or at least expression data should be grouped according to experiment. Second, results from the gMCS approach are not really comparable to results returned from GIMME since they basically return different things – networks returned from GIMME allow to directly determine essentiality of RRM1 while based on the gMCS approach the authors check whether for each data set there exists at least one gMCSs that only contains lowly expressed genes besides RRM1. Thus, one would already by chance expect that gMCSs more likely reports lethality (since for each data set there are 18 gMCSs that are tested).

Response: As required by both reviewers, we conducted the comparison of our approach with GIMME and iMAT grouping samples by cell lines. We should note here that, while we were grouping different samples into different cell lines, we realized that one sample was incorrectly assigned to JJN3, particularly JJN3-7 (GSM1374574) (see Supplementary Table 1 in the previous version of the manuscript). This sample was removed from our analysis. This incorrectly labeled sample (coming from gastric cancer) was the only one where our gMCS approach did not predict RRM1 as essential in the analysis presented in the previous version of the manuscript. In addition, as noted above, we included 4 samples of U266 cell line in our study. Overall, we considered in our study 21 samples: 7 samples of JJN3, 9 samples of H929, 1 sample of KMS-28-BM and 4 samples of U266. A summary of the results obtained at the sample and cell line level can be found in Supplementary Table 6. In the main manuscript we presented the results obtained at the level of each cell, in accordance with experimental validation, as suggested by both reviewers.

Our approach correctly predicted the essentiality of RRM1 in JJN3, H929 and KMS-28-BM, as well as the non-essentiality of RRM1 in U266. GIMME failed in H929 and U266, while iMAT in KMS-28-BM. Overall, these results show a superior performance of our approach than GIMME and iMAT. Again, given the results found, we would like to thank the comments made by both reviewers.

Supplementary Table 6: Prediction of the essentiality of RRM1 at the sample and cell line level in the MM study. Green coloring implies essentiality of RRM1, while red coloring non-essentiality.

GSM	gMCS	GIMME	iMAT	Cell Line	gMCS	GIMME	iMAT
GSM229051	Green	Green	Green	JJN3	Green	Green	Green
GSM915718	Green	Green	Red				
GSM915719	Green	Green	Red				
GSM915720	Green	Green	Red				
GSM1094684	Green	Red	Red				
GSM1094685	Green	Red	Red				
GSM1374579	Green	Red	Red				
GSM351746	Green	Red	Red	H929	Green	Red	Green
GSM451261	Green	Red	Red				
GSM451264	Green	Red	Red				
GSM451267	Green	Red	Green				
GSM511161	Green	Green	Green				
GSM511162	Green	Green	Red				
GSM511163	Green	Green	Green				
GSM562817	Green	Green	Red				
GSM662887	Green	Red	Red				
GSM887227	Green	Green	Red	KMS-28BM	Green	Green	Red
GSM363377	Red	Green	Green	U266	Red	Green	Red
GSM363399	Green	Green	Red				
GSM562821	Green	Green	Red				
GSM887721	Red	Green	Red				

On the other hand, we agree with the reviewer that iMAT and GiMME return different things. Certainly, for other purposes where metabolic reconstruction is necessary, GIMME and iMAT could be more informative. However, our analysis in this manuscript is focused on gene essentiality analysis and we can find a number of works in the literature where GiMME and iMAT are used for this purpose⁸. In fact, our algorithm, GiMME and iMAT use the same input data (human metabolic network and gene expression data) and, therefore, they can be unbiasedly compared to predict gene essentiality. With the results presented above, we think our approach is more accurate than GiMME and iMAT for predicting gene essentiality.

The reviewer should note that the number of gMCSs is actually not related with the chance of predicting lethality. Consider the example in the figure below. We have 3 gMCSs which explain the essentiality of g_4 for activating r_6 : $\{g_1, g_4\}$; $\{g_2, g_4\}$; $\{g_3, g_4\}$ (assuming one-to-one gene-reaction association). In this case, GIMME/iMAT would also obtain r_4 as essential, since the reconstruction in both cases leads to the removal of the pathway through r_1 , r_2 and r_3 .

The differences obtained between our approach and GIMME/iMAT comes from the bias introduced in the reconstruction process. This bias is different

depending on the algorithm used, as observed in the side-by-comparison discussed above (GiMME and iMAT obtains different outcomes). By avoiding the reconstruction step, where some lowly-expressed genes become active, we remove a source of bias and make our approach more accurate and sensitive to predict gene essentiality.

Minor points:

The coloring scheme in Figure 2A is misleading. If 5 is the cut-off for an enzyme to be expressed, this should be white. Otherwise it is quite difficult to grasp what particular shade of red actually indicates the cut-off for considering an enzyme as expressed.

Response: We thank the reviewer for his/her comments. We changed the coloring scheme of Figure 2A as suggested by the reviewer.

In l. 241 the authors state that they did not consider iMAT due to its high computational demand and GIMME providing better results in a previous test of several methods for building context-specific metabolic networks. There is some criticism concerning the fairness of these previous tests (Machado & Herrgard, 2014) including that they have been performed on prokaryotic metabolic networks (which are focused on growth and the validation data considering fluxes in growing bacteria) while methods for reconstructing context-specific metabolic networks have typically been developed for eukaryotic networks (where rapid growth typically is not an ultimate imperative). Moreover, from personal experience I know that depending on the choice of cut-offs applying iMAT typically requires only seconds and rarely in some severe cases (depending on the choice of parameters) probably a couple of hours for determining context-specific networks in Recon v2.04. Thus, it should be possible to build these networks for the just 18 samples the authors are considering and include iMAT in the comparison.

Response: As mentioned above, we included iMAT in our analysis. See Supplementary Note 2 for details of the computational implementation of iMAT. We thank the reviewer for this comment since the inclusion of iMAT certainly enriched the side-by-side comparison of our approach with existing methods in the literature.

In l. 265-270 the authors state that they used the gene with highest expression as the one most important for each gMCS. As there is no one-to-one relationship between abundance and importance of a gene, the authors should check whether they probably haven't always used the expression of the same gene as proxy for the expression strength of partner genes of RRM1. Beyond that it would also be more sensible to probably consider an aggregate of expression of partner genes of RRM1 and not just a single one.

Response: This comment is very interesting. We first checked whether we always used the expression of the same gene as proxy for the expression strength of partner genes of RRM1. This was not the case, as shown below for gMCS6. Certainly, the gene with highest expression differs across different cell lines (see Supplementary Data 1).

Samples	gMCS ₆
NCIH23-LUNG	SLC25A15
HCC70-BREAST	SDHB
EFO21-OVARY	SLC25A15
COV362-OVARY	SDHB
LAMA84-HAEMATOPOIETIC-AND-LYMPHOID-TISSUE	SLC25A15
TE10-OESOPHAGUS	ABAT
EFE184-ENDOMETRIUM	COASY
HCC2218-BREAST	SLC25A19
NCIH1299-LUNG	SLC25A15
JHOC5-OVARY	SLC25A15
EFM19-BREAST	ABAT
MONOMAC6-HAEMATOPOIETIC-AND-LYMPHOID-TISSUE	SLC25A19
AGS-STOMACH	ABAT
PANC0813-PANCREAS	SLC25A15
NCIH661-LUNG	SLC25A15
HT29-LARGE-INTESTINE	SLC25A15
NCIH1437-LUNG	COASY
GP2D-LARGE-INTESTINE	SLC25A15
PANC0327-PANCREAS	SLC25A15
BT474-BREAST	ABAT
SKCO1-LARGE-INTESTINE	SLC25A15
MIAPACA2-PANCREAS	SLC25A15
HCC1954-BREAST	SLC25A15
MDAMB453-BREAST	ALDH4A1
BT20-BREAST	ABAT
RKO-LARGE-INTESTINE	SLC25A15
MCF7-BREAST	ABAT
SNU840-OVARY	ABAT
HT55-LARGE-INTESTINE	ABAT
ZR7530-BREAST	ABAT

On the other hand, we repeated the same analysis using the average and sum of the expression of partner genes and the results are quite similar (Supplementary Table 7). In fact, we obtained even better p-values than using the gene with the maximum expression level, as shown below. As the most relevant results are kept, we maintain the results as were presented in the previous version of the manuscript

Supplementary Table 7. Summary of adjusted Spearman's correlation p-values in Achilles Scatter plots of different gMCSs using **max**, **mean** and **sum** of the expression of the partner genes of RRM1

	Achilles Scatter Plot – MAX		Achilles Scatter Plot - MEAN		Achilles Scatter Plot - SUM	
	p-value	adj. p-value	p-value	adj. p-value	p-value	adj. p-value
gMCS₁	0.7422	0.7624	0.7422	0.87315	0.7422	0.87315
gMCS₂	0.7624	0.7624	0.8914	0.89136	0.8914	0.89136
gMCS₃	0.7009	0.7624	0.8414	0.88566	0.8414	0.88566
gMCS₄	0.7009	0.7624	0.7894	0.8771	0.7894	0.8771
gMCS₅	0.0012	0.006	0.0012	0.00496	0.0012	0.00496
gMCS₆	0.0018	0.0071	0.004	0.01128	0.004	0.01128
gMCS₇	0.0953	0.1466	0.0151	0.03357	0.0151	0.03357
gMCS₈	0.0037	0.0122	0.0069	0.01727	0.0069	0.01727
gMCS₉	0.064	0.1067	0.1218	0.16238	0.1218	0.16238
gMCS₁₀	0.0004	0.0047	0.0006	0.004	0.0006	0.004
gMCS₁₁	0.1501	0.2001	0.1146	0.16238	0.1146	0.16238
gMCS₁₂	0.0318	0.0794	0.0039	0.01128	0.0039	0.01128
gMCS₁₃	0.0318	0.0794	0.1627	0.20338	0.1627	0.20338
gMCS₁₄	0.1919	0.2398	0.0361	0.0602	0.0361	0.0602
gMCS₁₅	0.0534	0.097	0.0210	0.0382	0.0210	0.0382
gMCS₁₆	0.117	0.1671	0.0205	0.0382	0.0205	0.0382
gMCS₁₇	0.0534	0.097	0.046	0.0707	0.046	0.0707
gMCS₁₈	0.0006	0.0047	0.0008	0.004	0.0008	0.004
gMCS₁₉	0.0472	0.097	0.0008	0.004	0.0008	0.004
gMCS₂₀	0.0007	0.0047	0.0004	0.004	0.0004	0.004

The authors state that they used all gene expression data originating from a specific chip for determining lowly expressed genes. Since gene expression in cancer cell lines is probably strongly different from gene expression in normal human tissue or cell lines, they should check how robust their results are if considering, for instance, just gene expression from cancer cell lines (e.g. NCI-60).

Response: The Gene Expression Barcode algorithm 3.0 (ref. 9) use thousands of samples of different tissues and conditions to build an expression call for each probe set in Affymetrix microarrays. Based on this expression call, lowly expressed genes are determined for each analyzed sample (see Methods section) and adapting Barcode to take into account only cancer samples is not possible in the available R package. While implementing an in-house system for such analysis is indeed interesting, it represents an effort that falls beyond the purpose on the current study. In any case, Barcode is sufficiently robust to capture cancer-specific expression patterns even with the inclusion of samples from normal tissues.

Reviewer #2 (Remarks to the Author):

Summary

In the manuscript by Apaolaza et al., the authors present a method that uses the concept of gene minimum cut sets (gMCSs) to predict and exploit synthetic lethality in cancer metabolism. They first introduce a novel method to expand the concept of minimum cut sets from the reaction level to the gene level. Then, they apply this method to explore the effect of knock down of the RRM1 gene in 18 different gene expression contexts/instances involving three cell lines of multiple myeloma. Their method predicts RRM1 to be essential in 17 of these 18 samples. Next, they validate experimentally the essentiality of RRM1 using shRNA in the three cell lines experiments. Finally, they report a correlation between the expression of the gMCSs group in 30 different cell lines and the ATARiS essentiality score of RRM1 measured in shRNA screens.

We believe that this is innovative work with a potentially promising route to personalized medicine and, especially, for the much needed patient stratification application. Nevertheless, we feel that the method presented here requires considerable additional experimental validation to establish its predictive power before we can support its publication in Nature Communications.

Response: We are sincerely grateful for the constructive comment of the reviewer. Undoubtedly, the comments of both reviewers helped us to show and establish the predictive power of our gMCS approach.

Major Comments

1. Further experimental validation of the method. In this work, the authors present a method that can be used to predict the essentiality of a gene in a certain context. This can be easily tested using publically available shRNA screens for multiple gene knockdowns. To establish the general predictive power of their method, the authors should prove the ability of this method to predict the essentiality of genes other than RRM1 in multiple cell lines using shRNA screens and the gene expression of these cell-lines. Only then, and if and only if, this approach may be of interest to the wide readership of Nature Communication. Note that this does not require new experiments! Just take a published collection of genome wide knockdowns of genes across many different cell-lines (e.g., like Achilles) and the associated cell-line transcriptomics, and overlay that expression data on the predicted gMCSs of the different genes, and show that you can predict their essentiality!...

Response: We would like to thank the reviewer for this important suggestion. In fact, a similar question was made by Reviewer 1, which was addressed in detail in the answer to her/his second comment. We summarize below the major changes introduced in the revised version of the manuscript.

First, we analyze the performance of our algorithm in a broader context and its capacity to identify other targets based on the Project Achilles data (Figure 1d). In addition, we provided further insights about RRM1 in MM. In particular, we identified a cell line in MM where our algorithm does not return RRM1 as essential: U266. We provided experimental validation of the non-essentiality of RRM1 in U266 (see new Figure 3). This shows that our algorithm is able to capture in which cases the inhibition of RRM1 is (or not) lethal in MM. Finally, in order to extend the conclusions attained for RRM1 in MM to other cancer types, we carried out additional experimental validation in H23, a lung cancer cell line used in Figure 4, where our algorithm predicted its dependence on RRM1 using gMCSs. Experimental results again confirmed our prediction (see Supplementary Figure 26). We think these results helped us to further show the predictive power of our gMCS approach, as required by the reviewer.

2. As the shRNA validation of the RRM1 gene essentiality is performed in cell lines and not samples, it would be preferred if the authors would predict essentiality in a cell line specific manner.

Response: As suggested by the reviewer, we presented in the main text the results obtained when the essentiality of RRM1 is predicted at the cell line level.

To establish the clinical relevance of this method, the authors could show that the expression level of the gMCS groups that they find is also relevant in clinical data, e.g. in predicting patient survival in the TCGA cohort (its quite straightforward to test).

Response: This is indeed an interesting suggestion as the analysis of clinical relevance of the results presented is the next logical step in our study. In order to perform a survival analysis, we used a public microarray datasets of MM patients, which involves 328 samples¹⁰, including survival data for 279 patients¹¹. Expression levels were normalized using the Gene Expression Barcode algorithm 3.0 (ref. 1). The survival analysis of MM patients according to RRM1 expression indicated that those patients with higher RRM1 expression have a worse overall survival. This result is consistent with recent studies¹².

Next, we focused on gMCS₆, since it was the only gMCS being significant in the two different statistical analyses accomplished at the cell line level (see the first comment of Reviewer 1). We conducted a survival analysis according to the expression of the genes involved in gMCS₆ (using its maximum and average expression) but we did not find any significant result. This was partially expected since identified gMCSs can be used as molecular markers of the response to RRM1 inhibition, but not as a surrogate for prognosis, unless specific inhibitors of RRM1 were to be used, which is not the case. In other words, the prognostic value from the gMCS would only be such, if we could inhibit RRM1.

Minor Comments

1. Please check the writing style so that methods are all described in the methods section and the results are listed subsequent order in the results section.

Response: As suggested by the reviewer, we moved all technical information describing the advance brought by our approach to the Supplementary Material (see page X). In our opinion, the Results and Methods sections are now better organized.

2. Please provide statistical estimates for the reported finding. E.g in line 201 to line 203, there should be a p-value representing the significance of these results.

Response: The comment raised by the reviewer is pertinent. We have provided additional statistical evidence of many of the results observed. See "Statistical Analyses" in Methods section.

In particular, for the reported finding in lines 201-203, we calculated an adjusted p-value using a one-sided Binomial test namely to evaluate whether its frequency in the MM samples considered is statistically significant. See "Statistical Analyses" in Methods section.

References

1. Cowley, G. S. *et al.* Parallel genome-scale loss of function screens in 216 cancer cell lines for the identification of context-specific genetic dependencies. *Scientific Data* **1**, 140035– (2014).
2. Hart, T., Brown, K. R., Sircoulomb, F., Rottapel, R. & Moffat, J. Measuring error rates in genomic perturbation screens: gold standards for human functional genomics. *Molecular systems biology* **10**, 733– (2014).
3. Barretina, J. *et al.* The Cancer Cell Line Encyclopedia enables predictive modelling of anticancer drug sensitivity. *Nature* **483**, 603–307 (2012).
4. Shao, D. D. *et al.* ATARiS: Computational quantification of gene suppression phenotypes from multisample RNAi screens. *Genome Research* **23**, 665–678 (2012).
5. Becker, S. A. & Palsson, B. O. Context-Specific Metabolic Networks Are Consistent with Experiments. *PLOS Computational Biology* **4**, e1000082– (2008).
6. Shlomi, T., Cabili, M. N., Herrgard, M. J., Palsson, B. O. & Ruppin, E. Network-based prediction of human tissue-specific metabolism. *Nat Biotech* **26**, 1003–1010 (2008).
7. Tobalina, L., Pey, J., Rezola, A. & Planes, F. J. Assessment of FBA Based Gene Essentiality Analysis in Cancer with a Fast Context-Specific Network Reconstruction Method. *PLOS ONE* **11**, e0154583– (2016).
8. Pacheco, M. P., Pfau, T. & Sauter, T. Benchmarking Procedures for High-Throughput Context Specific Reconstruction Algorithms. *Frontiers in Physiology* **6**, 410– (2015).
9. McCall, M. N. *et al.* The Gene Expression Barcode 3.0: improved data processing and mining tools. *Nucleic acids research* **42**, 938–43 (2014).
10. Terragna, C. *et al.* Correlation between eight-gene expression profiling and response to therapy of newly diagnosed multiple myeloma patients treated with thalidomide-dexamethasone incorporated into double autologous transplantation. *Annals of Hematology* **92**, 1271–1280 (2013).
11. Kuiper, R. *et al.* A gene expression signature for high-risk multiple myeloma. *Leukemia* **26**, 2406–2413 (2012).
12. Sagawa, M. *et al.* Ribonucleotide reductase large subunit (RRM1) as a novel therapeutic target in multiple myeloma. *Clinical Cancer Research* (2017).doi:10.1158/1078-0432.CCR-17-0263

REVIEWERS' COMMENTS:

Reviewer #1 (Remarks to the Author):

The authors addressed all of my concerns convincingly.

Reviewer #2 (Remarks to the Author):

We enjoyed reading the authors responses to our questions and feel this revision is considerably stronger and that our concerns were adequately addressed.

To clarify our second comment from the original review (about survival analysis), we would expect to see the level of expression of the SL partners matter only when RRM1 is lowly expressed. Low expression of RRM1 can be thought of as a proxy of RRM1 inhibition, which is likely why the expression level of RRM1 is significant for the survival analysis. However, low expression (or inhibition) of RRM1 alone is not sufficient on its own (or the SL partners wouldn't matter). Therefore, incorporating the predicted SL partners should strengthen the survival predictions (for the patients who lowly express RRM1).

Response to the Reviewer's comments (manuscript NCOMMS-16-29429A)

We thank the positive comments of both reviewers. Again, we would like to express our gratitude to the reviewers for their excellent suggestions on the previous version of the manuscript.

Below you can find the detailed answer to the issue raised by Reviewer 2.

Reviewer #1

The authors addressed all of my concerns convincingly.

Response: We are very grateful since the quality of the manuscript substantially improved following the reviewer's advice.

Reviewer #2 (Remarks to the Author):

We enjoyed reading the authors responses to our questions and feel this revision is considerably stronger and that our concerns were adequately addressed.

Response: As with Reviewer 1, we are very grateful because, thanks to the reviewer's advice, a stronger version of the manuscript was submitted.

To clarify our second comment from the original review (about survival analysis), we would expect to see the level of expression of the SL partners matter only when RRM1 is lowly expressed. Low expression of RRM1 can be thought of as a proxy of RRM1 inhibition, which is likely why the expression level of RRM1 is significant for the survival analysis. However, low expression (or inhibition) of RRM1 alone is not sufficient on its own (or the SL partners wouldn't matter). Therefore, incorporating the predicted SL partners should strengthen the survival predictions (for the patients who lowly express RRM1).

Response: As suggested by the reviewer, we conducted the survival analysis for MM patients with low RRM1 expression (*i.e.* we filtered MM patients with high RRM1 expression) based on the expression of SL partners in gMCS₆. As in the previous response letter, we used a public microarray dataset of MM patients, which involves 328 samples¹, including survival data for 279 patients². Expression levels were normalized using the Gene Expression Barcode algorithm 3.0 (ref. 3).

The Kaplan-Meier Overall Survival (OS) curve can be observed in the figure below. In line with the suggestion of the reviewer, patients with a higher expression of SL partners show a worse OS (p -value= 0.014, Cox proportional hazard model), which strengthens the survival predictions.

Even with this positive result, in this article we show that the expression of SL partners in gMCS₆ can be used as a molecular marker of the response to RRM1 inhibition, but not as a surrogate for prognosis, unless specific inhibitors of RRM1 were to be used, which is not the case. In other words, the prognostic value from the gMCS would only be such, if we could inhibit RRM1.

We understand the assumption made by the reviewer (certainly interesting): patients with low RRM1 expression can be used as a proxy for RRM1 inhibition. However, as MM treatments regulate different molecular targets, we prefer to be cautious and leave a more detailed clinical study as a future work.

References

1. Terragna, C. *et al.* Correlation between eight-gene expression profiling and response to therapy of newly diagnosed multiple myeloma patients treated with thalidomide-dexamethasone incorporated into double autologous transplantation. *Annals of Hematology* **92**, 1271–1280 (2013).
2. Kuiper, R. *et al.* A gene expression signature for high-risk multiple myeloma. *Leukemia* **26**, 2406–2413 (2012).
3. McCall, M. N. *et al.* The Gene Expression Barcode 3.0: improved data processing and mining tools. *Nucleic acids research* **42**, 938–43 (2014).